# PERCEPTUAL CONTEXT AND SENSITIVITY IN IMAGE QUALITY ASSESSMENT: A HUMAN-CENTRIC APPROACH

## ABSTRACT

Blind Image Quality Assessment (BIQA) mirrors subjective made by human observers. The Human Visual System (HVS) assesses image quality by combining a global perspective of the contrasting relationships among samples of varying quality with a local analysis of individual images. However, current BIQA methodologies tend to emphasize local evaluations but overlook the contrasting relationship inherent in global perception, leading to the incomprehensive representation of human subjective assessment. Consequently, the representation learning of the BIQA model remains suboptimal. To address this, we present the Perceptual Context and Sensitivity in BIQA (CSIQA), a novel metric learning paradigm that seamlessly integrates efficient human-centric global and local evaluations into the BIQA methodology. Specifically, the CSIQA comprises two primary components: 1). A Quality Context Contrastive Learning module, that is equipped with different contrastive learning strategies to effectively capture potential quality correlations in the **global context** of the dataset. 2). A Quality-aware mask attention module, which employs the random masking mechanism to ensure the consistency with visual **local sensitivity**, thereby improving the model's perception of local distortions. Extensive experiments on eight standard BIQA datasets demonstrate the superior performance to the state-of-the-art BIQA methods, *i.e.,* achieving the PLCC values of 0.941 (↑ 3.3% vs. 0.908 in TID2013) and 0.920 (↑ 2.6% vs. 0.894 in LIVEC). Our code is available at https://anonymous.4open.science/r/CSIQA-9627/.

## 1 INTRODUCTION

*Image Quality Assessment* (IQA) (Madhusudana et al., 2022; Ding et al., 2020; Gu et al., 2020) is devoted to estimating the perceptive quality of a digital image consistent with the human visual system (HVS). It has been applied in many computer vision pieces of research, such as image restoration (Banham & Katsaggelos, 1997) and super-resolution (Dong et al., 2015). IQA methods are generally classified into three types based on the availability of reference images, i.e. full-reference (Cheon et al., 2021), reduced-reference (Wang et al., 2016), and no-reference or blind IQA (BIQA) (Wu et al., 2020). In real-world scenarios, it is common to encounter situations where reference images are unavailable. Consequently, BIQA that does not require the reference image becomes more appealing and applicable. However, two challenges remain in the existing BIQA:

- **1) Context.** Existing methods typically train deep learning models by treating each sample as independent. However, for a more accurate assessment of quality, it's crucial to consider the contrastive relationships between different quality samples (Fig. 1(b)), since humans are better adept at learning optimal image quality representations through the process of comparing the quality of different images (Ponomarenko et al., 2009; Liu et al., 2017).

- **2) Sensitivity.** Current BIQA methods utilize domain knowledge from large datasets like ImageNet (Deng et al., 2009) to learn about image quality representation (Fig. 1(a)). However, the semantic features extracted from pre-trained models are sub-optimal for BIQA since BIQA predominantly depends on distortion-sensitive attributes to perceive the visual quality of pictures with different semantic contexts (Su et al., 2020; Zhang et al., 2023).

Many state-of-the-art (SOTA) BIQA methods have been proposed to address these two challenges. For instance, Su et al. (2020) depends on pre-trained models to understand image content and then

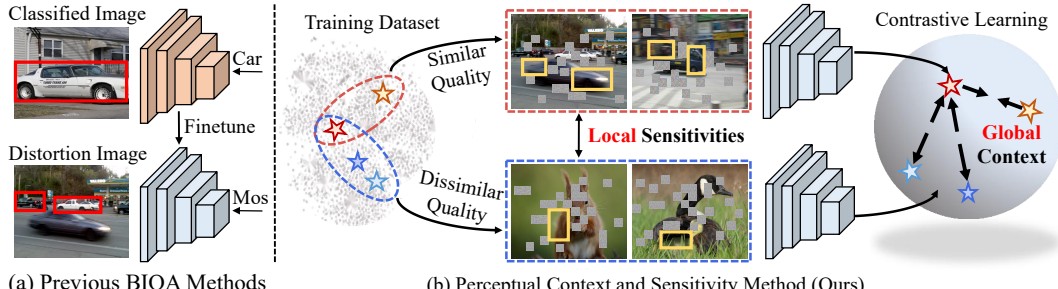

(a) Previous BIQA Methods        (b) Perceptual Context and Sensitivity Method (Ours)

Figure 1: The illustration of the difference between (a) previous BIQA methods and (b) our CSIQA. Previous BIQA methods predominantly rely on semantic features to directly learn image quality, neglecting the inherent **Global context contrastive** and **Local sensitivities** of the HVS. In CSIQA, we fully explore the above two properties by using contrastive learning across images in the training data and quality-aware masked attention, which obtain comprehensive quality-aware features.

judge the image quality through a hypernetwork, but it does not consider the quality relevance of the context of the dataset (*i.e.,* Challenge **1**)). Recent advancements, such as Rankiqa (Liu et al., 2017) and CONTRIQUE (Madhusudana et al., 2022), have explored global quality relations (*i.e.,* Challenge **2**)). However, these methods have their limitations. Rankiqa, for instance, utilizes rank learning which can be theoretically vulnerable to IQA data noise (Theorem. 2), affecting optimal model learning. CONTRIQUE, while harnessing contrastive learning, does not fully exploit the contrast information available from authentic images of similar quality, which limits its generalization ability in authentic scenes. To this end, we introduce the Perceptual Context and Sensitivity in BIQA (CSIQA), a novel approach that fully leverages quality ground truths to enhance the model's discernment capabilities in authentic datasets.

To address the Challenge **1**), we introduce the **Quality Context Contrastive Learning** approach, inspired by HVS proficiency in comparing image quality. This approach constructs easy positive and negative sets based on quality-level similarities. To mitigate overfitting, we incorporate a **Hard Sampling Strategy** within a knowledge distillation framework. This strategy involves sampling hard positive (negative) samples that have similar (different) quality but different (similar) pseudo-labels generated by pre-trained teacher reasoning. By focusing on these challenging samples, our model is better equipped to prioritize difficult predictions and utilize sample information effectively. We then employ InfoNCE (Oord et al., 2018) to capture the global properties in the embedding space, which reflect the inherent structure of the training data, resulting in more precise quality predictions.

To address the Challenge **2**), we propose the **Quality-Aware Mask Attention** method that enforces models focusing on patch-level localized distortions rather than image-level semantic information. Specifically, we design a new quality-aware (QUA) token to help the model extract low-level quality features that are detached from high-level semantic information. In the forward process, the quality-aware token interacts almost the same way as the Class (CLS) token, except that it only focuses on visible patches. This mechanism is incorporable with quality context contrastive learning to achieve better quality perception, which facilitates more effective quality-related information exploration. We summarize the contributions of this work as follows:

• We identify the limitations of existing Blind Image Quality Assessment (BIQA) methods. In contrast to previous approaches that solely concentrate on single-image analysis, we delve into exploring and analyzing the cross-image context quality correlation as a prior, which offers valuable insights for enhancing the accuracy of quality prediction.

• Inspired by the HVS, we propose a novel quality context contrastive learning that leverages abundant quality-aware contrastive information in the training dataset. This paradigm offers a well-structured quality embedding space to enhance the model's discrimination of quality features.

• A novel quality-aware mask attention mechanism is introduced to combine contrastive learning for improving quality perception. By randomly masking patches, the designed quality-aware token can effectively emphasize low-level quality features that complement high-level semantic information.

• Our approach integrates smoothly into existing BIQA networks without altering the base model or adding testing burdens (Tab. 2). Experiments on eight IQA benchmarks show that CSIQA outperforms similar algorithms, proving our method's effectiveness and efficiency.

## 2 RELATED WORK

**IQA with Local Perspective.** The early BIQA method (Zeng et al., 2018; Liu et al., 2017) was based on the convolutional neural network (CNN) thanks to its powerful feature expression ability. CNN-based BIQA methods, such as (Zhang et al., 2018; Su et al., 2020), often pre-train for object recognition and then fine-tune for IQA. While these pre-trained features are somewhat aligned with quality-aware features(Su et al., 2020), the transition from classification to IQA remains challenging due to the abstract nature of classification features. Recently, Vision Transformer (ViT)(Dosovitskiy et al., 2021) methods have emerged in BIQA, with two main architectures: hybrid transformer(Golestaneh et al., 2022) and pure transformer (Ke et al., 2021; You & Korhonen, 2021). The hybrid approach merges CNNs for local features with transformers for long-range features. Pure ViT-based methods depend on the CLS which originally for image content description, focuses on higher-level visual abstractions, but the CLS token's primary design for semantic perception limits its BIQA effectiveness.

**IQA with a global quality contrastive Perspective.** In IQA, ranking-based learning is commonly used to model global quality relationships. Specifically, Zhang et al. (2018) employed discrete ranking data from images with consistent content but varying distortion levels for quality prediction. On the other hand, Zhang et al. (2021) used continuous ranking data based on *Mean Opinion Scores* (MOS) and differences in subjective ratings. Research by (Ma et al., 2017b; 2019) incorporated binary ranking data from FR-IQA methods in their training. However, their reliance on reference images limits the performance in authentic distortions datasets. Notably, many current rank-based IQA methods, such as (Liu et al., 2017; Ma et al., 2017a), require manual hyperparameter selection, making them more sensitive to noisy data perception. Inspired by the success of contrastive learning, Madhusudana et al. (2022) explored the use of it for quality representation. However, this approach struggles to handle the quality contrast relationship in real images with complex distortion types. In this paper, we improve the sample screening process by leveraging ground truth and enhancing local sensitivity, aiming to better capture image perceptual features.

## 3 METHODOLOGY

### 3.1 OVERVIEW

In this paper, we introduce the *perceptual Context and Sensitivity IQA* (CSIQA) designed to predict image quality from both global and local perspectives. As depicted in Fig. 2, CSIQA seamlessly integrates two main components: Global Quality Context Contrastive Learning and Local Quality-Aware Mask Attention. Initially, CSIQA divides the input image into patches, which are then fed into the Mask transformer encoder. After processing through $H$ layers of Quality-Aware Mask Attention (§ 3.2), we obtain the **local** sensitive feature $\boldsymbol{F}^h$ at the $h$-th layer. Once $\boldsymbol{F}^h$ is acquired, it is promptly passed to the Quality Context Contrastive Learning module (§ 3.3). Here, $\boldsymbol{F}^h$ and its corresponding positive and negative samples are used as inputs for the InfoNCE to model the **global** contrastive relation. The selection of simple and hard positive and negative samples (§ 3.4) is based on the similarity between ground truths and pseudo-labels. Finally, the output $\boldsymbol{F}^H$ from the final encoder layer is refined further through a transformer decoder to derive the quality score.

### 3.2 QUALITY-AWARE MASK TRANSFORMER

The self-attention of the transformer (Dosovitskiy et al., 2021) is effective in comprehensively representing perceptual features. The global semantic features extracted by a pre-trained upstream backbone may not be optimal for capturing quality awareness, as HVS is highly sensitive to local distortions even when the overall image quality is good (Larson & Chandler, 2010; Su et al., 2020). Therefore, we propose the use of random mask block interactions with learnable tokens, which enable the learning of local quality-aware features that are independent of global semantic information.

Let $\boldsymbol{F} = \{\boldsymbol{F}_{qua}; [\boldsymbol{F}_{cls}; \boldsymbol{F}_{1:N-1}]\} \in \mathbb{R}^{(N+1)\times D}$ be the embedding sequence, where $N$ is the number of patches plus one CLS token, $D$ is the embedding dimension, $\boldsymbol{F}_{qua}$ is the learnable quality-aware token. Three linear projection layers transform $\boldsymbol{F}$ into matrices $\boldsymbol{Q}, \boldsymbol{K}, \boldsymbol{V}$ for query, key, and value. To regulate the interaction between $\boldsymbol{F}_{qua}$ and the other tokens, we construct a mask $\boldsymbol{m} \in \mathbb{R}^{(N+1)}$.

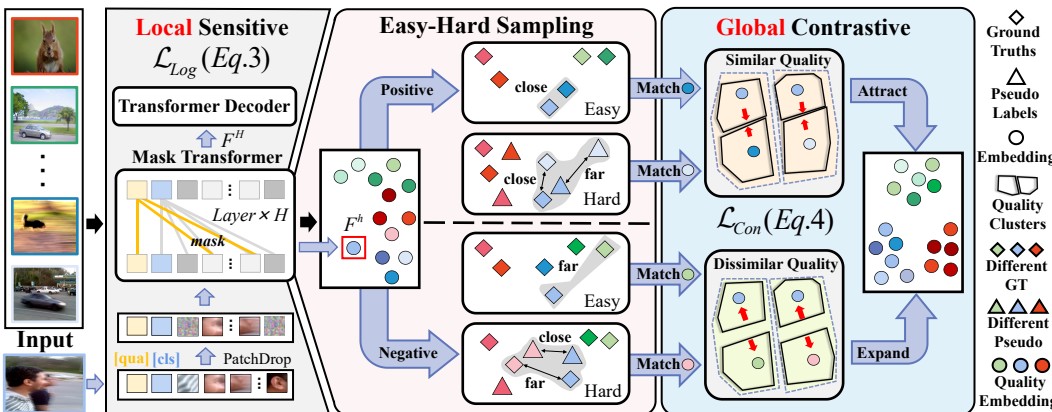

Figure 2: The overview of CSIQA, in which similar colors represent similar qualities. CSIQA first obtains each layer's image features through **Local** Quality-Aware Mask Attention (§ 3.2). After that, features are used to compute contrastive loss in the **Global** Quality Context Contrastive Learning (§ 3.3). Positive and negative sample selection is based on ground truth and pseudo-label similarity (§ 3.4). The final quality score is derived by refining the feature via a transformer decoder.

To maintain the original attention, our attention mask $M$ is constructed as follows:

$$m_i = \begin{cases} 1 - \mathbb{1}\{ \text{ the } i\text{-th patch is masked } \}, & i < N \\ 1, & i = N \end{cases}, \quad M = \begin{bmatrix} \mathbf{1}^{N \times N} & \mathbf{0}^{N \times 1} \\ m^{1 \times N} & 1 \end{bmatrix}, \quad (1)$$

where $\mathbb{1}\{\cdot\}$ is the indicator function, the probability of the random mask is 0.3. Intuitively, Eq. 1 means that $\boldsymbol{F}_{qua}$ only attends to the patch tokens that are not randomly masked. Suppose the Encoder has $H$ attention layers, the Multi-Head Self-Attention (MHSA) is utilized, as follows:

$$\mathbf{F}^h = \begin{cases} \sigma\left( \log \boldsymbol{M} + \mathbf{Q}^h \left(\mathbf{K}^h\right)^\top \right) \mathbf{V}^h + \mathbf{F}^{h-1}, & 0 < h \leq H \\ \mathbf{F}', & h = 0 \end{cases}, \quad (2)$$

where $\boldsymbol{Q}^h, \boldsymbol{K}^h, \boldsymbol{V}^h$ are linear transformations of $\boldsymbol{F}^{h-1}$, $\sigma$ is Softmax function. Finally, the encoder produces output features after $H$ layers. To provide meaningful interpretations for the CLS and QUA tokens, a transformer decoder is employed (Qin et al., 2023). This approach enhances the significance of the extracted features for image quality. During training, the smooth $\mathcal{L}_1$ loss is minimized. The introduction of quality-aware mask attention improves the learning capability of quality-aware features in the transformer-based BIQA model, resulting in enhanced prediction accuracy and generalization ability.

### 3.3 SUPERVISED CONTRASTIVE IMAGE QUALITY ASSESSMENT

**Logits-Based Supervised Loss.** Knowledge distillation (KD) can improve performance by regularizing the Logits-based knowledge in the teacher model (Wang et al., 2019). To leverage this technique to IQA, we supervise our student model using both ground truth and pre-trained teacher-generated pseudo-labels. In order to indicate the effectiveness of this technique, we give detailed proof in Theorem 1. This analysis confirms that the expected loss with our method is notably reduced when contrasted with training that relies solely on ground truth. As a result, the reliability of the pseudo-labels is affirmed which facilitates rapid convergence and acts as a safeguard against overfitting. Given the teacher network and our student network produce mappings $Y_T$ and $Y_S$ for the image $\boldsymbol{I}$ to quality scores, respectively. Given $\hat{Y}$ as the ground truth label for image $\boldsymbol{I}$, $\lambda_1$ are the hyperparameters, $\|\cdot\|_1$ represents the $\ell_1$ regression loss, the Logits-Based Supervised loss defined as follows:

$$\mathcal{L}_{\text{Log}} = \| \hat{Y} - Y_S \|_1 + \lambda_1 \| Y_T - Y_S \|_1, \quad (3)$$

**Quality Context Contrastive Learning.** Inspired by HVS, humans are better at making accurate quality predictions when presented with contrast. The logits-based loss functions focus solely on local

intra-image analysis, neglecting the essential quality contrast global information between different images within the training dataset. In IQA, rank learning can be employed to model the global quality relationships across the dataset. However, rank learning heavily relies on the accuracy of rankings, which becomes challenging to achieve when comparing similar images (Gao et al., 2015). As per Theorem 2, the model is significantly influenced by the noise in ranking order. Our analysis suggests that this is primarily due to the use of randomly sampled samples for comparison without any selective filtering. Consequently, we introduce a quality context contrastive learning approach that optimally utilizes label information to selectively filter specific samples for comparison, enhancing the quality of learned embeddings. We define quality context Contrastive loss as follows:

$$\mathcal{L}_{\text{Con}} = \frac{1}{|\mathbb{F}_{\text{pos}}|} \sum_{\boldsymbol{F}^+ \in \mathbb{F}_{\text{pos}}} -\log \frac{\exp(\boldsymbol{F} \cdot \boldsymbol{F}^+/\tau)}{\exp(\boldsymbol{F} \cdot \boldsymbol{F}^+/\tau) + \sum_{F^- \in \mathbb{F}_{\text{neg}}} \exp(\boldsymbol{F} \cdot \boldsymbol{F}^-/\tau)}. \tag{4}$$

In the framework of our proposed contrastive learning approach, the symbols $\mathbb{F}_{\text{pos}}$ and $\mathbb{F}_{\text{neg}}$ are employed to denote the sets of image embeddings for positive and negative samples, respectively. Our method capitalizes on quality labels, enabling a discerning identification and selection of pivotal samples for comparison, thereby refining the embedding space. The overarching objective is to diminish the distance between samples with analogous quality while expanding the distance for those with divergent quality. This methodology bolsters the precision and robustness of the embeddings.

**Overall Loss.** The logits-based supervised loss in Eq. 3 and the contrastive loss in Eq. 4 benefit each other during training. The $\mathcal{L}_{\text{Log}}$ helps the student network identify relevant quality features. The $\mathcal{L}_{\text{Con}}$ refines the embedding space, promoting compactness for similar-quality images and separability for different-quality ones. This approach aligns with human perception from a global to local attentive evaluation. Here, the hyperparameter $\lambda_2$ are used to balance $\mathcal{L}_{\text{Log}}$ and $\mathcal{L}_{\text{Con}}$, the selection of it can be found in the Appendix E, $H$ denote the number of encoder layers. The total loss is defined as:

$$\mathcal{L}_{\text{Scon}} = \sum_I (\mathcal{L}_{\text{Log}} + \lambda_2 \sum_{h=1}^{H} \mathcal{L}_{\text{Con}}). \tag{5}$$

### 3.4 EASY-HARD SAMPLING STRATEGY FOR CONTRASTIVE LEARNING

**Define Positive-Negative Samples.** To distinguish between positive and negative samples, we begin by extending the score vector $\boldsymbol{y} \in \mathbb{R}^{B \times 1}$ to a matrix $\boldsymbol{Y} \in \mathbb{R}^{B \times B}$, where $B$ represents the batch size. The score distance matrix $\boldsymbol{Y}_d$ is then calculated using the Manhattan distance $\mathcal{D}$. For each pair of images $\boldsymbol{I}_a$ and $\boldsymbol{I}_j$, we sort the $a$-$th$ row of $\boldsymbol{Y}_d$ to obtain the ascending distance vector $\hat{\boldsymbol{Y}}_d[a:]$. The corresponding index of the score distance between $\boldsymbol{I}_a$ and $\boldsymbol{I}_j$ in $\hat{\boldsymbol{Y}}_d[a:]$ is denoted as $C$. If the index $C$ is less than the product of $\gamma_1$ and $B$, it is a positive sample. Conversely, if it is greater than the product of $\gamma_2$ and $B$, it is a negative sample. Here, $\gamma_1$ and $\gamma_2$ are hyperparameters that we empirically set as $20\%$ and $60\%$ of the batch size. The definition of the classifier is as follows:

$$\boldsymbol{Y}_d = \mathcal{D}(\boldsymbol{Y} - \boldsymbol{Y}^T), \quad \text{Classifier}(\boldsymbol{I_a}, \boldsymbol{I_j}) \in \begin{cases} \mathbb{F}_{\text{pos}}, & \text{if } C \leq \gamma_1 \cdot B, \\ \mathbb{F}_{\text{neg}}, & \text{if } C \geq \gamma_2 \cdot B. \end{cases} \tag{6}$$

**Easy-Hard Example Sampling.** Previous studies (Schroff et al., 2015; Kalantidis et al., 2020) have highlighted the importance of selecting training samples in metric learning. When considering image embedding represented by $\boldsymbol{F}$, the gradient of the contrastive loss (*Eq. 4*) can be expressed as follows:

$$\frac{\partial \mathcal{L}_{\text{Con}}}{\partial \boldsymbol{F}} = -\frac{1}{\tau |\mathbb{F}_{\text{pos}}|} \sum_{F^+ \in \mathbb{F}_{\text{pos}}} \left( (1 - p^+) \cdot \boldsymbol{F}^+ - \sum_{F^- \in \mathbb{F}_{\text{neg}}} p^- \cdot \boldsymbol{F}^- \right). \tag{7}$$

The matching probability between a positive or negative sample $\boldsymbol{F}^{+/-}$ and the embedding $\boldsymbol{F}$ represents as $p^{+/-} = \frac{\exp(\boldsymbol{F} \cdot \boldsymbol{F}^{+/-}/\tau)}{\sum_{\boldsymbol{F}' \in \mathbb{F}_{\text{pos}} \cup \mathbb{F}_{\text{neg}}} \exp(\boldsymbol{F} \cdot \boldsymbol{F}'/\tau)} \in [0, 1]$. Hard negative and positive samples (*i.e.,* negatives but similar features and positives but dissimilar features) have a higher contribution to the gradient compared to easy negative and positive samples (Wang et al., 2021). Learning from hard samples helps the model acquire more quality-aware features. To identify Hard samples, we propose utilizing a distillation framework for sampling. However, to mitigate the risk of the model getting trapped in local optima, we also include easy samples in the sampling process.

- **Easy Example Sampling.** The sampling process is similar to the method described in Eq. 6. For each feature embedding $\boldsymbol{F}$, we sample the top 10% and the last 40% of available samples from the $\boldsymbol{Y_d}$ matrix sorted in ascending order. as easy positive and negative samples, respectively.
- **Hard Example Sampling.** Our hard sampling strategy uses the relative error between the teacher network's predicted scores and the ground truth for sampling. We generate a pseudo-scores vector $\hat{\boldsymbol{y}}$ using the teacher network, expanded into $\hat{\boldsymbol{Y}}$. The $\hat{\boldsymbol{Y_d}}$ matrix is computed similarly to Eq. 6. The final error matrix is $\boldsymbol{Y}_{\text{error}} = \frac{\boldsymbol{Y_d}}{\hat{\boldsymbol{Y_d}}}$. We select the top 10% and bottom 40% samples from the sorted $\boldsymbol{Y}_{\text{error}}$ matrix, intersecting with samples delineated in Eq. 6 to reduce sampling deviation.

## 3.5 THEORETICAL ANALYSIS

**Theorem 1** *(Reliability of $\mathcal{L}_{\text{Log}}$) Let $T(x)$ denote the output from the teacher model, $R(x)$ be the ground truth, and $S(x)$ be the student's output for an input image $x$. Define the weights $a = \frac{1}{1+\lambda}$ and $b = \frac{\lambda}{1+\lambda}$, where $a, b \in [0,1]$ and $a + b = 1$.The objective function $\hat{S}(x)$ is then defined as:*

$$\hat{S}(x) = \underset{S}{\operatorname{argmin}} \sum_i a \left| S\left(x_i\right) - T\left(x_i\right) \right| + b \left| S\left(x_i\right) - R\left(x_i\right) \right|. \tag{8}$$

*From this, we derive the expected value of the optimal loss function. Here,$\lambda$ is a hyperparameter that balances the contributions of the teacher's output and the ground truth, the simplified result is:*

$$\mathbb{E}_x \left( \left| \hat{S}(x) - T(x) \right| + \lambda \left| \hat{S}(x) - R(x) \right| \right) < (1 + \lambda) \mathbb{E}_x \left( \left| R(x) - T(x) \right| \right). \tag{9}$$

*Proof.* Please refer to the Appendix for complete proof. A.1

Theorem 1 demonstrates that the anticipated loss when using the proposed Logits-Based Supervised method is lower than when training exclusively with the ground truth. This underscores the reliability of the pseudo-labels generated using our approach.

**Theorem 2** *(Robustness of $\mathcal{L}_{\text{Con}}$) Given the formalisms of contrastive learning (**CL**) and rank learning (**RL**), we define their losses as:*

$$\mathcal{L}_{RL} = -|y_1 - y_0|, \quad \mathcal{L}_{CL} = \min |y_+ - y| - \max |y_- - y|,$$

*where $y$ is an image label, and $y_+$ and $y_-$ are its positive and negative samples. Given a label $\hat{y}_0$ for a sample point $\hat{x}_0$ and $n + 2$ samples $\{(\hat{x}_1, \hat{y}_1), ..., (\hat{x}_{n+2}, \hat{y}_{n+2})\}$, if the probability of $|\hat{y}_i - \hat{y}_0| \geqslant |\hat{y}_j - \hat{y}_0|$ is $p$, then the probability that all samples are farther from $\hat{y}_0$ than the positive sample $\hat{y}_j$ is $p^{n+1}$. The associated probability density is given by $(n + 1) \cdot p^n \cdot h(\eta, \xi)$, where $h(\eta, \xi)$ denotes the probability density of $p$ with respect to $\eta$ and $\xi$.*

*The robustness $\Delta_{RL}$ of noise on rank learning can be expressed as:*

$$\int \left| [p^n + (1-p^n)] |\eta + \xi| \cdot h - [\hat{p}^n + (1-\hat{p})^n] |\xi| \cdot \hat{h} \right| \cdot f \cdot g \cdot (n+1) \, d\eta d\xi.$$

*The robustness $\Delta_{CL}$ of noise on contrastive learning can be expressed as:*

$$\left| \int -|\eta + \xi| f \cdot g d\eta d\xi - \int -|\xi| f \cdot g d\eta d\xi \right| = \int ||\eta + \xi| - |\xi|| f \cdot g d\eta d\xi$$

*In order to compare the robustness $\Delta_{CL}$ and $\Delta_{RL}$ of rank learning and contrastive learning:*

$$\tilde{p} = \underset{\{p,\hat{p}\}}{\operatorname{argmax}} \{[p^n + (1-p)^n], [\hat{p}^n + (1-\hat{p})^n]\}, \quad \tilde{h} = \underset{\{h,\hat{h}\}}{\operatorname{argmax}} \{h, \hat{h}\},$$

$$\Delta_{CL} \leq \int ||\eta + \xi| - |\xi|| \cdot [\tilde{p}^n + (1-\tilde{p})^n] \cdot \tilde{h} \cdot f \cdot g \cdot (n+1) \, d\eta d\xi \leq \Delta_{RL}.$$

*Proof.* Please refer to the Appendix A.2 for comprehensive proof.

From the Theorem 2, we infer that for large $n$ and when $0 < \tilde{p} < 1$, **CL** exhibits greater robustness to noise $\eta$ due to its inherent filtering mechanism. Pseudo-labels from the teacher can introduce noise, which might alter the ranking in **RL**, potentially misleading the model. For instance, it might change the relation "a < b" to "b < a". However, the sensitivity of **CL** to such noise is mitigated, especially when there are numerous negative samples. Given that **CL** typically uses multiple negative samples for each positive one, a few noisy negatives exert minimal influence on the overall learning process.

Table 1: Performance comparison measured by averages of SRCC and PLCC, where bold entries indicate best results, underlines indicate the second-best. CO. stands for the CONTRIQUE method.

| Method | LIVE PLCC | LIVE SRCC | CSIQ PLCC | CSIQ SRCC | TID2013 PLCC | TID2013 SRCC | KADID PLCC | KADID SRCC | LIVEC PLCC | LIVEC SRCC | KonIQ PLCC | KonIQ SRCC | LIVEFB PLCC | LIVEFB SRCC | SPAQ PLCC | SPAQ SRCC |
|---|---|---|---|---|---|---|---|---|---|---|---|---|---|---|---|---|
| BRISQUE (Mittal et al., 2012) | 0.944 | 0.929 | 0.748 | 0.812 | 0.571 | 0.626 | 0.567 | 0.528 | 0.629 | 0.629 | 0.685 | 0.681 | 0.341 | 0.303 | 0.817 | 0.809 |
| ILNIQE (Zhang et al., 2015) | 0.906 | 0.902 | 0.865 | 0.822 | 0.648 | 0.521 | 0.558 | 0.534 | 0.508 | 0.508 | 0.537 | 0.523 | 0.332 | 0.294 | 0.712 | 0.713 |
| BIECON (Kim & Lee, 2016) | 0.961 | 0.958 | 0.823 | 0.815 | 0.762 | 0.717 | 0.648 | 0.623 | 0.613 | 0.613 | 0.654 | 0.651 | 0.428 | 0.407 | - | - |
| MEON (Ma et al., 2017b) | 0.955 | 0.951 | 0.864 | 0.852 | 0.824 | 0.808 | 0.691 | 0.604 | 0.710 | 0.697 | 0.628 | 0.611 | 0.394 | 0.365 | - | - |
| WaDIQaM (Bosse et al., 2017) | 0.955 | 0.960 | 0.844 | 0.852 | 0.855 | 0.835 | 0.752 | 0.739 | 0.671 | 0.682 | 0.807 | 0.804 | 0.467 | 0.455 | - | - |
| DBCNN (Zhang et al., 2018) | 0.971 | 0.968 | 0.959 | 0.946 | 0.865 | 0.816 | 0.856 | 0.851 | 0.869 | 0.851 | 0.884 | 0.875 | 0.551 | 0.545 | 0.915 | 0.911 |
| TIQA (You & Korhonen, 2021) | 0.965 | 0.949 | 0.838 | 0.825 | 0.858 | 0.846 | 0.855 | 0.85 | 0.861 | 0.845 | 0.903 | 0.892 | 0.581 | 0.541 | - | - |
| MetaIQA (Zhu et al., 2020) | 0.959 | 0.960 | 0.908 | 0.899 | 0.868 | 0.856 | 0.775 | 0.762 | 0.802 | 0.835 | 0.856 | 0.887 | 0.507 | 0.54 | - | - |
| P2P-BM (Ying et al., 2020) | 0.958 | 0.959 | 0.902 | 0.899 | 0.856 | 0.862 | 0.849 | 0.84 | 0.842 | 0.844 | 0.885 | 0.872 | 0.598 | 0.526 | - | - |
| HyperIQA (Su et al., 2020) | 0.966 | 0.962 | 0.942 | 0.923 | 0.858 | 0.840 | 0.845 | 0.852 | 0.882 | 0.859 | 0.917 | 0.906 | 0.602 | 0.544 | 0.915 | 0.911 |
| TReS (Golestaneh et al., 2022) | 0.968 | 0.969 | 0.942 | 0.922 | 0.883 | 0.863 | 0.858 | 0.859 | 0.877 | 0.846 | 0.928 | 0.915 | 0.625 | 0.554 | - | - |
| MUSIQ (Ke et al., 2021) | 0.911 | 0.940 | 0.893 | 0.871 | 0.815 | 0.773 | 0.872 | 0.875 | 0.746 | 0.702 | 0.928 | 0.916 | 0.661 | 0.566 | 0.921 | 0.918 |
| DACNN (Pan et al., 2022) | 0.980 | 0.978 | 0.957 | 0.943 | 0.889 | 0.871 | 0.905 | 0.905 | 0.884 | 0.866 | 0.912 | 0.901 | - | - | 0.921 | 0.915 |
| CO. (Madhusudana et al., 2022) | 0.961 | 0.960 | 0.955 | 0.942 | 0.857 | 0.843 | **0.937** | **0.934** | 0.857 | 0.845 | 0.906 | 0.894 | 0.641 | 0.580 | 0.919 | 0.914 |
| DEIQT (Qin et al., 2023) | 0.982 | 0.980 | 0.963 | 0.946 | 0.908 | 0.892 | 0.887 | 0.889 | 0.894 | 0.875 | 0.934 | 0.921 | 0.663 | 0.571 | 0.923 | 0.919 |
| CSIQA (Ours) | **0.985** | **0.983** | **0.970** | **0.960** | **0.941** | **0.926** | 0.919 | 0.919 | **0.920** | 0.898 | **0.943** | 0.929 | **0.688** | **0.594** | **0.935** | **0.930** |

Table 2: Integration of quality context contrastive learning into IQA networks TIQA and DEIQT, both employing the VIT backbone. The "+" symbol indicates the incorporation of our learning strategy.

| Method | CSIQ PLCC | CSIQ SRCC | LIVEC PLCC | LIVEC SRCC | KonIQ PLCC | KonIQ SRCC |
|---|---|---|---|---|---|---|
| TIQA | 0.945 | 0.930 | 0.881 | 0.863 | 0.930 | 0.914 |
| TIQA + | $\textbf{0.963}_{+1.8\%}$ | $\textbf{0.951}_{+2.1\%}$ | $\textbf{0.910}_{+2.9\%}$ | $\textbf{0.883}_{+2.0\%}$ | $\textbf{0.940}_{+1.0\%}$ | $\textbf{0.926}_{+1.2\%}$ |
| DEIQT | 0.961 | 0.950 | 0.894 | 0.875 | 0.934 | 0.921 |
| DEIQT + | $\textbf{0.968}_{+0.7\%}$ | $\textbf{0.956}_{+0.6\%}$ | $\textbf{0.916}_{+2.2\%}$ | $\textbf{0.893}_{+1.8\%}$ | $\textbf{0.941}_{+0.7\%}$ | $\textbf{0.926}_{+0.5\%}$ |

# 4 EXPERIMENTS

## 4.1 EXPERIMENT SETUP

**Benchmark Datasets and Evaluation Protocols.** Our method is evaluated on eight public BIQA datasets. including four synthetic datasets, namely LIVE (Sheikh et al., 2006), CSIQ (Larson & Chandler, 2010), TID2013 (Ponomarenko et al., 2015), and KADID (Lin et al., 2019), and four authentic datasets, namely LIVEC (Ghadiyaram & Bovik, 2015), KONIQ (Hosu et al., 2020), LIVEFB (Ying et al., 2020), and SPAQ (Fang et al., 2020). The authentic datasets feature images from various mobile devices captured by different photographers, while the synthetic datasets contain images distorted using methods like Gaussian blur. Details of these datasets, including their sizes and distortion types, can be found in the Appendix C. Performance is assessed using Spearman's Order Correlation Coefficient (SRCC) and Pearson's Linear Correlation Coefficient (PLCC), both ranging from 0 to 1, with higher values denoting superior accuracy and monotonicity.

**Implementation Details.** Following (Qin et al., 2023; Ke et al., 2021), we employ a pre-trained encoder based on ViT-S (Touvron et al., 2022), with a depth of 12, and the Decoder depth is set to one. We train the model for 9 epochs using a learning rate of $2 \times 10^{-4}$ and a decay factor of 10 every 3 epochs. The size of the batch depends on the size of the dataset, ranging from 16 to 128. We split the datasets into 80% for training and 20% for testing and repeat the process ten times to reduce performance bias. We report the average of SRCC and PLCC to quantify the model's performance in terms of prediction accuracy and monotonicity. More details about the student and teacher network's structure, training, performance, and selection principle can be found in Appendix C.

## 4.2 COMPARISON WITH SOTA BIQA METHODS

The results of the comparison between CSIQA and 15 classical or state-of-the-art BIQA methods, which include hand-crafted feature-based BIQA methods like ILNIQE (Zhang et al., 2015) and BRISQUE (Mittal et al., 2012), as well as deep learning-based methods such as MUSIQ (Ke et al., 2021) and MetaIQA (Zhu et al., 2020), are presented in Tab. 1. It can be observed across the eight datasets that CSIQA outperforms all other methods in terms of performance. Achieving leading performance on all of these datasets is a challenging task due to the wide range of image content and distortion types. Therefore, these observations confirm the effectiveness and superiority of CSIQA in accurately characterizing image quality.

Table 3: SRCC on the cross datasets validation. The best performances are highlighted in boldface.

| Training | Testing | DBCNN | P2PBM | HyperIQA | TReS | CO. | DEIQT | CSIQA |
|----------|---------|-------|-------|----------|------|-----|-------|-------|
| LIVEFB | KonIQ | 0.716 | 0.755 | 0.758 | 0.713 | - | 0.733 | **0.760** |
| LIVEFB | LIVEC | 0.724 | 0.738 | 0.735 | 0.740 | - | 0.781 | **0.787** |
| LIVEC | KonIQ | 0.754 | 0.740 | 0.772 | 0.733 | 0.676 | 0.744 | **0.777** |
| KonIQ | LIVEC | 0.755 | 0.770 | 0.785 | 0.786 | 0.731 | 0.794 | **0.814** |
| LIVE | CSIQ | 0.758 | 0.712 | 0.744 | 0.761 | **0.823** | 0.781 | 0.818 |
| CSIQ | LIVE | 0.877 | - | 0.926 | - | 0.925 | 0.932 | **0.945** |

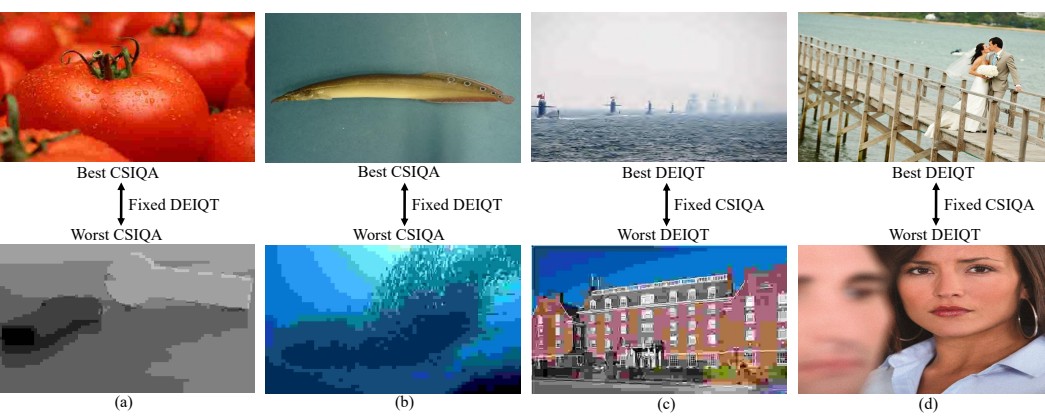

Figure 3: gMAD competition results. (a) Fixed DEIQT at low quality. (b) Fixed DEIQT at high quality. (c) Fixed CSIQA at low quality. (d) Fixed CSIQA at high quality.

**Generalization Capability Validation. 1)** To assess the generalization capacity of CSIQA, we perform cross-dataset validation experiments where the model is trained on one dataset and tested on others, without any fine-tuning or parameter adjustment. Tab. 3 displays the results, presenting the average SRCC obtained across five datasets. It is worth noting that CSIQA outperforms the SOTA models in all four cross-authentic dataset experiments, showcasing significant improvements. **2)** To further assess our framework's generalization, we trained models on the entire LIVE database and then tested them using the gMAD competition (Ma et al., 2016b) on the Waterloo Database (Ma et al., 2016a). As shown in Fig. 3, our CSIQA remains stable in defense, with minimal changes in perceived image quality. However, as an attacker, it selects image pairs with significant quality differences, showcasing its strength in both roles. Further details are in the Appendix F.

**Framework Portability.** To demonstrate the generalization of our proposed quality contrastive learning framework (only contrastive learning and distillation are included), which can be applied to any IQA model. In Tab. 2, we perform an ablation study on previous methods TIQA and DEIQT which with the backbone of VIT for fine-tuning. We generate pseudo-labels using the original models (performance shown in rows one and three) and then apply our proposed contrastive framework. The reported results indicate the general efficacy of our method in enhancing existing IQA networks.

### 4.3 ABLATION STUDY

**Effect of Logit Distillation.** In Tab.4, we juxtapose indices *a)* with *d)*, *f)* with *i)*, and *g)* with *k)* to elucidate the varying enhancements brought about by logit distillation. By integrating noisy pseudo-labels during the training phase, we bolster the universal inference capabilities for quality estimation, thereby reinforcing generalization. This observation aligns with and validates Theorem1.

**Effect of Quality-Aware Mask Attention.** Next, we compare index *b)* and *c)*, the mask attention((abbr. $MA$) improves quality representation extraction, yielding a performance boost of 0.5% to 1% across all datasets. Comparing index *c)* and *d)*, the $MA$ enhances the distillation process by providing more effective supervision information. Moreover, the best performance is achieved after utilizing the $MA$ module when other modules are considered ( *i)* v.s. *k)* ). These results underscore the improvement in image quality perception from mask encoder optimization in Appendix D.

**Effect of Quality-aware Contrast with Easy-hard Sampling.** *Context Contrastive Learning:* We juxtapose indices *a)* and *e)* to gauge the influence of contrastive learning. Notably, employing contrastive learning with easy sampling (denoted as $EA$) yields a marked enhancement, registering

Table 4: Ablation experiments on three datasets. Here, $EA$ and $HA$ refers to contrastive learning with easy and easy-hard sampling; $MA$ refers to mask attention; $LD$ refers to logits distillation.

| Index | $EA$ | $HA$ | $MA$ | $LD$ | TID2013 | | LIVEC | | KonIQ | |
|---|---|---|---|---|---|---|---|---|---|---|
| | | | | | PLCC | SRCC | PLCC | SRCC | PLCC | SRCC |
| a) | | | | | 0.910 | 0.891 | 0.899 | 0.878 | 0.930 | 0.914 |
| b) | | | ✔ | | 0.918 | 0.898 | 0.906 | 0.879 | 0.934 | 0.919 |
| c) | | | | ✔ | 0.922 | 0.903 | 0.908 | 0.885 | 0.935 | 0.921 |
| d) | | | ✔ | ✔ | $0.927_{+1.7\%}$ | $0.907_{+1.6\%}$ | $0.910_{+1.1\%}$ | $0.882_{+0.4\%}$ | $0.938_{+0.8\%}$ | $0.924_{+1.0\%}$ |
| e) | ✔ | | | | 0.927 | 0.909 | 0.908 | 0.882 | 0.935 | 0.919 |
| f) | ✔ | ✔ | | | 0.939 | 0.925 | 0.910 | 0.889 | 0.936 | 0.919 |
| g) | ✔ | ✔ | ✔ | | $0.940_{+3.0\%}$ | $\mathbf{0.926}_{+3.5\%}$ | $0.912_{+1.3\%}$ | $0.889_{+1.1\%}$ | $0.938_{+0.8\%}$ | $0.923_{+0.9\%}$ |
| h) | ✔ | | ✔ | ✔ | 0.928 | 0.910 | 0.918 | 0.897 | 0.942 | 0.927 |
| i) | ✔ | ✔ | | ✔ | 0.940 | 0.925 | 0.914 | 0.892 | 0.941 | 0.926 |
| k) | ✔ | ✔ | ✔ | ✔ | $\mathbf{0.941}_{+3.1\%}$ | $\mathbf{0.926}_{+3.5\%}$ | $\mathbf{0.920}_{+2.1\%}$ | $\mathbf{0.898}_{+2.0\%}$ | $\mathbf{0.943}_{+1.3\%}$ | $\mathbf{0.929}_{+1.5\%}$ |

Figure 4: Visual comparisons between DEIQT and CSIQA. Scores in this figure represent MOS. CSIQA is well sensitive to local distortion, resulting in more accurate quality prediction.

an approximate 1.7% performance uptick relative to the baseline on TID2013. *Incorporation of Hard Sampling:* When amalgamated with hard sampling (denoted as $HA$), an additional increment nearing 1% is discerned, with sporadic enhancements manifesting across other datasets. Such disparities in advancements across datasets can be ascribed to the delineation of positive and negative samples, compounded by the batch sizes chosen for training. For instance, the batch sizes of 16 and 128, designated for training on the LIVEC and KonIQ datasets respectively, might induce either a paucity or a surfeit of positive and negative samples. More details can be found in the Appendix F.

**Visualization of attention map.** Using GradCAM (Selvaraju et al., 2017), we visualize the feature attention map in Fig.4. CSIQA adeptly concentrates on local distortions (e.g., motion blur in column three, underexposure in the last, and overexposure in the inverted second column.). In contrast, DEIQT often misdirects its focus to clear backgrounds, overlooking distortions like exposure or motion blur. Fig.4 contrasts the quality predictions of CSIQA and DEIQT on authentic distortion images. CSIQA consistently surpasses DEIQT, especially excelling in moderately distorted images. Additional visualizations are provided in the supplementary Appendix F.

## 5 CONCLUSION

In this study, we present CSIQA, an innovative quality contrastive learning methodology for BIQA. Contrary to conventional models that predominantly concentrate on local intra-image analysis, CSIQA accentuates both the global context quality contrast and the nuances of local distortions. It incorporates a quality context contrastive module, facilitated by pseudo-labels, to underscore samples with unpredictable quality, thereby capturing global quality contrastive relations. Additionally, a quality-aware mask attention module is employed to refine quality discrimination by focusing on local distortions. Experiments show that CSIQA surpasses current SOTA methods. Moreover, our training approach seamlessly integrates with existing methods, yielding notable enhancements.

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

APPENDIX

This Appendix contains five sections. § A.1 offers a theoretical proof underscoring the reliability of incorporating pseudo-labels. § A.2 provides a theoretical demonstration that contrastive learning exhibits greater robustness to noise compared to rank learning. § B presents some details of the CSIQA model architecture § C delves deeper into the training and evaluation specifics. § D extends our discussion with additional ablation studies related to our CSIQA. This includes an exploration of the quality-aware token and an examination of the efficacy of logit distillation in hastening convergence. § E conducts a sensitivity analysis focusing on hyperparameters, specifically addressing mask probability and the weight of the loss function. § F presents a thorough qualitative assessment of CSIQA. This encompasses visual representations of feature distributions across a range of quality scores and a visualization of the activation map.

# A PROOFS

## A.1 QUALITATIVE PROOF OF PROPOSITION 1

The soundness of pseudo-labels is meticulously ensured by both mathematical proofs and experiments. We start with the following three assumptions. 1) The teacher network generates enough pseudo-labels. 2) The pseudo-labels generated by the teacher network are relatively balanced with the samples in the dataset. 3) The teacher randomly samples in the process of generating pseudo-labels

**1. Definition:** $T(x)$ represents the teacher's distribution. $R(x)$ represents the true distribution. $S(x)$ represents the student's distribution. $f(x)$ is the probability distribution of $x$.

**2. Minimization of Expression:** Let $a, b \in [0, 1]$, and $a + b = 1$. The goal is to find the student's distribution $\hat{S}(x)$ that minimizes the expression involving the differences between the student's distribution and the teacher's distribution, as well as the true distribution:

$$\hat{S}(x) = \underset{S}{\operatorname{argmin}} \sum_i a \left| S(x_i) - T(x_i) \right| + b \left| S(x_i) - R(x_i) \right|$$

**3. Inequality Derivation:** The optimal solution, denoted as $\hat{S}(x)$, was pursued within the context of this analysis. The derivation is as follows:

$$\int \left( a \left| \hat{S}(x) - T(x) \right| + b \left| \hat{S}(x) - R(x) \right| \right) f(x) dx$$

$$\leq \int (a|S - T| + b|S - R|) f(x) dx$$

$$(\text{Let: } S(x) = bT + aR)$$

$$\int \left( a \left| \hat{S}(x) - T(x) \right| + b \left| \hat{S}(x) - R(x) \right| \right) f(x) dx$$

$$\leq \int (a|bT + aR - T| + b|bT + aR - R|) f(x) dx$$

$$= \int (a^2|R - T| + b^2|T - R|) f(x) dx$$

$$= (a^2 + b^2) \int |R - T| f(x) dx$$

$$< (a + b) \int |R - T| f(x) dx$$

**4. Conclusion:** Through deduction, the $\hat{S}(x)$ will be influenced by the teacher and will be superior to the teacher. We conclude that with a sufficiently abundant quantity of pseudo-labels (ensuring the expectation), and a relatively balanced distribution between pseudo-labels and dataset samples (for any values of 'a' and 'b'), our anticipated loss is lower compared to training solely with ground truth. Consequently, under these three assumptions, the reliability of pseudo-labels can be ensured.

## A.2 QUALITATIVE PROOF OF PROPOSITION 2

**1. The formalism of CL and Rank Learning:** We present the general formalism of CL and rank learning at the label level: The loss function of CL simplifies to "distance of positive pair-distance of negative pair" (here we disregard the "partition function"). For the purpose of demonstration, we select one sample each from the maximum and minimum values, using them as negative and positive examples respectively. The proof for multiple examples follows the same logic.:

$$\mathcal{L}_{CL} = \min |y_+ - y| - \max |y_- - y|$$

For a label $y$ sample, $y_+$ and $y_-$ are its positive/negative samples. The general form of loss in RL is typically the "negative of the distance":

$$\mathcal{L}_{RL} = -|y_1 - y_0|$$

**2. Rank Learning (RL) Analysis:** Consider a label $y_0$ for a sample point. We sample around $y_0$ using a distribution to obtain $y_1$, where $y_1 = y_0 + \xi$ and $\xi$ follows the sampling distribution. Given noise $\eta$ in the "Teacher" model, we define the noisy label as $z = y_1 + \xi + \eta$ when incorporating noise $\eta$. The RL with noise $\eta$ can be expressed as:

$$\int -|z - y_0| f(\eta) g(\xi) d\eta d\xi = \int -|\eta + \xi| f(\eta) g(\xi) d\eta d\xi$$

In the absence of noise $\eta$ in RL:

$$\int -|y_1 - y_0| g(\xi) d\xi = \int -|\xi| f(\eta) g(\xi) d\eta d\xi$$

**3. Contrastive Learning (CL) Analysis:** With $n + 2$ samples $\{\hat{y}_1, ..., \hat{y}_{n+2}\}$, the probability of $|\hat{y}_i - y_0| \geq |\hat{y}_j - y_0|$ is $p$. Thus, the probability that all samples are farther from $y_0$ than the positive sample $\hat{y}_j$ is $p^{n+1}$. The probability density is $(n+1)p^n h(\eta, \xi)$, where $h(\eta, \xi)$ represents the probability density of $p$ concerning $\eta$ and $\xi$. The CL with noise $\eta$ can be expressed as:

$$\int (n+1) p^n h |\eta + \xi| fg d\eta d\xi - \int (n+1)(1-p)^n (-h) |\eta + \xi| fg d\eta d\xi$$

$$= \int (n+1) [p^n + (1-p)^n] |\eta + \xi| fgh d\eta d\xi$$

In the absence of noise $\eta$ in CL (due to the absence of noise $\eta$, $\hat{p}$ and $\hat{h}$ are used instead to replace $p$ and $h$ from before).

$$\int (n+1) \hat{p}^n \hat{h} |\xi| g d\xi - \int (n+1)(1-\hat{p})^n (-\hat{h}) |\xi| g d\xi$$

$$= \int (n+1) [\hat{p}^n + (1-\hat{p})^n] \hat{h} |\xi| fg d\eta d\xi$$

**4. Comparative Analysis of Robustness:** We qualitatively prove the robustness of CL and RL:

Evaluating the impact of noise on RL:

$$\left| \int -|\eta + \xi| fg d\eta d\xi - \int -|\xi| fg d\eta d\xi \right| = \int ||\eta + \xi| - |\xi|| fg d\eta d\xi \tag{10}$$

Evaluating the impact of noise on CL:

$$\left| (n+1) \{ \int [p^n + (1-p)^n] \cdot h |\eta + \xi| fg d\eta d\xi - \int [\hat{p}^n + (1-\hat{p})^n] \cdot \hat{h} |\xi| fg d\eta d\xi \} \right|$$

$$= \int |[p^n + (1-p^n)] \cdot |\eta + \xi| \cdot h - [\hat{p}^n + (1-\hat{p})^n] \cdot |\xi| \cdot \hat{h}| \cdot f \cdot g \cdot (n+1) d\eta d\xi \tag{11}$$

Let $\tilde{p} = \max\{p^n + (1-p)^n, \hat{p}^n + (1-\hat{p})^n\}$ and $\tilde{h} = \underset{\{h, \hat{h}\}}{\mathrm{argmax}}\{h, \hat{h}\}$, *i.e.,*

$$\text{Equ.11} \leq \int ||\eta + \xi| - |\xi|| \cdot [\tilde{p}^n + (1-\tilde{p})^n] \cdot \tilde{h} \cdot f \cdot g \cdot (n+1) d\eta d\xi \leq \text{Equ.10} \tag{12}$$

**5. Conclusion:** From the aforementioned equation, it can be deduced that when $n$ is sufficiently large and $0 < \tilde{p} < 1$, contrastive learning exhibits enhanced robustness to noise $\eta$ due to its inherent filtering mechanism. In our empirical validation, both contrastive learning and rank learning techniques were applied across various backbones, with training conducted on noisy labels. The experimental findings underscore that contrastive learning is notably more robust. Further details can be found in Section D.

## B    ARCHITECTURE

### B.1    ENCODER DETAILS

The quality-aware token in the encoder functions in a manner akin to the CLS token, leveraging a self-attention mechanism. Distinctively, it attends solely to visible patches. In the encoder's formulation (Equation 2 as depicted in the manuscript), a mask operation is designed to regulate the interaction scope of the quality token during forward propagation. The feature values from the encoder are subsequently refined through a Multi-Layer Perceptron (MLP) and a residual connection. The mathematical representation is as follows:

$$\mathbf{F}^h = \begin{cases} \sigma \left( \log \boldsymbol{M} + \mathbf{Q}^h \left( \mathbf{K}^h \right)^\top \right) \mathbf{V}^h + \mathbf{F}^{h-1}, & 0 < h \leq H \\ \mathbf{F}', & h = 0 \end{cases}$$

$$\boldsymbol{F}^h = \text{MLP} \left( \text{Norm} \left( \boldsymbol{F}^h \right) \right) + \boldsymbol{F}^h$$

$\boldsymbol{F} = \{\boldsymbol{F}_{qua}; [\boldsymbol{F}_{cls}; \boldsymbol{F}_{1:N-1}]\} \in \mathbb{R}^{(N+1) \times D}$ be the embedding sequence, where $N$ is the number of patches plus one CLS token. Here, $F_{qua}$ and $F_{cls}$ symbolize the QUA token and CLS token.

### B.2    DECODER DETAILS

Within the decoder, both the QUA token and CLS token undergo refinement. The QUA token is processed inside the Multi-Head Self-Attention (MHSA) block to discern dependencies among its constituents. The output from the MHSA is then integrated with residual connections, resulting in a query for the transformer decoder. This process is mathematically captured as:

$$\boldsymbol{Q}_{\text{qua}} = \text{MHSA} \left( \text{Norm} \left( \boldsymbol{F}_{\text{qua}} \right) \right) + \boldsymbol{F}_{\text{qua}}$$

Subsequently, tokens excluding the qua and CLS tokens are employed as keys and values for the decoder, represented by $\boldsymbol{K}_d$ and $\boldsymbol{V}_d$. This aids in deriving the refined features $F_{\text{qua}}$. The ultimate score, $Y$, is derived from the Multi-Head Cross-Attention (MHCA) as:

$$F_{\text{qua}} = \text{MLP} \left( \text{MHCA} \left( \text{Norm} \left( Q_{\text{qua}} \right), \boldsymbol{K}_d, \boldsymbol{V}_d \right) + Q_{\text{qua}} \right)$$

$$Y = \text{MLP} \left( \text{Cat} \left( \boldsymbol{F}_{\text{qua}}, \boldsymbol{F}_{\text{cls}} \right) \right)$$

## C    TRAINING AND EVALUATION DETAILS

**Student Network.** **1) Training.** To train the student network, we adopt a standard approach of randomly cropping input images into ten patches, each with a $224 \times 224$ resolution. Subsequently, we reshape these patches into a sequence of smaller patches with a patch size of p = 16 and an input token dimension of D = 384. Furthermore, we present additional training preprocessing details for various datasets, which are not included in the main paper, in Tab. 7. For different benchmarks, we employ different training settings for a fair comparison. Throughout the training process, we implement easy-hard sampling with a batch size of the dataset and compute the contrastive loss using InfoNCE (Oord et al., 2018). However, it is worth noting that during inference, both our quality context contrastive module and quality-aware mask attention module will be deprecated.

**2) Testing.** The number of training epochs is set to 9 and we report the results for the last epoch since the official data split does not have a validation set. It is worth noting that we do not make any adjustments to the hyperparameters during training and testing with different train-test dataset splits. and repeated this process 10 times to mitigate the performance bias and the average of SRCC and PLCC were reported.

**Teacher Network.** We will start with structure, training, performance, and selection principles. Let's introduce the Teacher network:

• **Structure.** The pre-trained teacher transformer consists of VIT-S (Touvron et al., 2022) and one decoder layer (denoted as "VIT-D"). It is worth noting that, as a general framework (Tab. 2), the teacher network is all the pre-trained version of the student network.

• **Training.** The training process of the teacher network is consistent with the training process of the student network, we use a learning rate of $2 \times 10^{-4}$ and a decay factor of 10 every 3 epochs to train the model for 9 epochs. The teacher training process has the same data set and test set as the student training, which ensures the independence and fairness of the content. During training and inference, we adopt a standard approach of randomly cropping input images into ten patches, each with a $224 \times 224$ resolution. Subsequently, we reshape these patches into a sequence of smaller patches with a patch size of p = 16 and an input token dimension of D = 384.

Table 5: Teacher performance on authentic and synthetic datasets.

| | LIVE | | CSIQ | | TID2013 | | KADID | |
|---|---|---|---|---|---|---|---|---|
| Method | PLCC | SRCC | PLCC | SRCC | PLCC | SRCC | PLCC | SRCC |
| Teacher | 0.978 | 0.978 | 0.960 | 0.945 | 0.910 | 0.891 | 0.904 | 0.902 |
| | LIVEC | | KonIQ | | LIVEFB | | SPAQ | |
| Method | PLCC | SRCC | PLCC | SRCC | PLCC | SRCC | PLCC | SRCC |
| Teacher | 0.900 | 0.878 | 0.930 | 0.914 | 0.667 | 0.567 | 0.922 | 0.920 |

• **Performance.** We guarantee a high accuracy of the pre-trained teacher model on multiple datasets to provide reliable pseudo-labels, as shown in Tab. 5. We further illustrate through experiments that our teacher model produces relatively reliable pseudo-labels: We compare the student model's performance using pseudo-labels with its performance using the true labels. Tab. 6 indicates that the student model achieves similar performance with both, thus the pseudo-labels are reliable.

Table 6: Experiments on the reliability ablation of teacher-generated pseudo-labels.

| | LIVE | | LIVEC | |
|---|---|---|---|---|
| Method | PLCC | SRCC | PLCC | SRCC |
| baseline baseline | 0.970 | 0.967 | 0.899 | 0.878 |
| CSIQA w/ pesudo | 0.983(+1.3%) | 0.982(+1.5%) | 0.916(+1.7%) | 0.898(+2.0%) |
| CSIQA w/ pesudo+label | 0.985(+1.5%) | 0.983(+2.1%) | 0.920(+2.1%) | 0.898(+2.0%) |

• **Selection principle.** In our experiments on the LIVEC dataset, we explored three distinct settings to understand the influence of teacher models on the student's performance: 1) Teacher models with identical architecture but varying performances. Specifically, we considered models VIT-D and VIT-DF. 2) Teacher models with differing architectures but consistent performance. For this, we evaluated models DBCNN (Zhang et al., 2018), HyperNet (Su et al., 2020), and VIT-D. 3) Teacher models with both different architectures and performances. We took into account models DBCNN, HyperNet, and VIT-DF. Here, VIT-DF represents the fully converged VIT-D. The PLCC of the remaining teachers is 0.863, 0.867, and 0.868.

Table 7: Training preprocessing details of selected BIQA datasets.

| Dataset | Resolution | Resize | Batch Size | Label Range |
|---|---|---|---|---|
| LIVE (Sheikh et al., 2006) | $768 \times 512$ | $512 \times 384$ | 12 | DMOS [0,100] |
| CSIQ (Larson & Chandler, 2010) | $512 \times 512$ | $512 \times 512$ | 12 | DMOS [0,1] |
| TID2013 (Ponomarenko et al., 2015) | $512 \times 384$ | $512 \times 384$ | 48 | MOS [0,9] |
| KADID (Lin et al., 2019) | $512 \times 384$ | $512 \times 384$ | 128 | MOS [1,5] |
| LIVEC (Ghadiyaram & Bovik, 2015) | $500P \sim 640P$ | $500P \sim 640P$ | 16 | MOS [1,100] |
| KonIQ (Hosu et al., 2020) | $768P$ | $512 \times 384$ | 128 | MOS [0,5] |
| LIVEFB (Ying et al., 2020) | $160P \sim 700P$ | $512 \times 512$ | 128 | MOS [0,100] |
| SPAQ (Fang et al., 2020) | $1080P \sim 4368P$ | $512 \times 384$ | 128 | MOS [0,100] |

Table 8: Experiments on the selection principle of teacher models.

| Teacher | DBCNN | | HyperNet | | VIT-D | | VIT-DF | |
|---|---|---|---|---|---|---|---|---|
| | PLCC | SRCC | PLCC | SRCC | PLCC | SRCC | PLCC | SRCC |
| baseline | 0.899 | 0.878 | 0.899 | 0.878 | 0.899 | 0.878 | 0.899 | 0.878 |
| CSIQA | **0.905** | **0.880** | **0.903** | **0.878** | **0.910** | **0.882** | **0.920** | **0.898** |

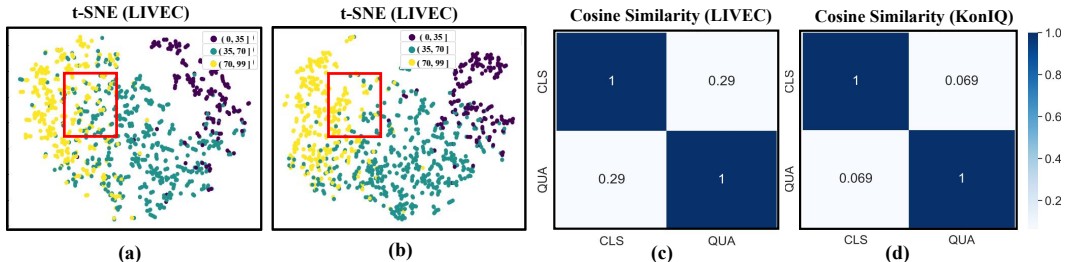

Figure 5: (a) and (b) depict the embedding space without and with the inclusion of a quality token. The comparison indicates that introducing the quality token in contrastive learning optimizes more quality-related features by incorporating semantic information. This results in a well-structured quality-aware embedding space (b). Furthermore, (c) and (d) demonstrate that the minimal similarity between the quality token and cls token highlights their distinct focuses on different regions of distortion. Specifically, the cls token prioritizes global distortion, while the quality token prioritizes local distortion. As a result, the quality perception becomes more comprehensive.

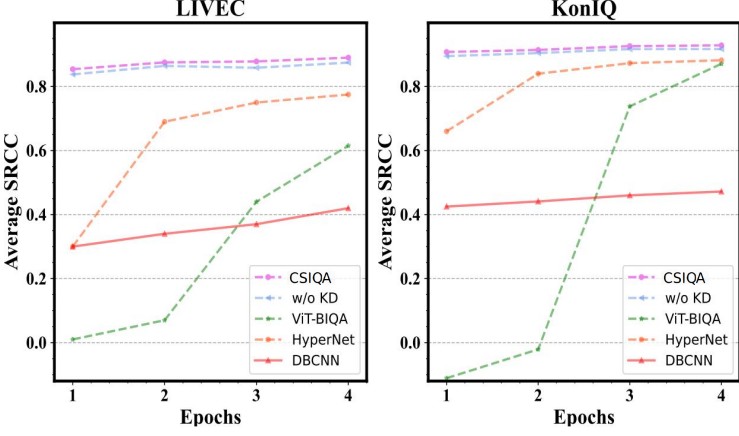

Figure 6: Average SRCC versus Epochs on different datasets ablation on logit distillation.

# D MORE ABLATION STUDIES

**Ablation about Quality-aware token.** To illustrate the effectiveness of introducing the quality token in generating a well-structured embedding space for contrastive learning, we categorize the quality scores of the LIVEC dataset into three distinct categories: heavily distorted images (score 0-35), moderately distorted images (score 35-70), and lightly distorted images (score 70-100). Each set includes a minimum of 300 images from the LIVEC real-world dataset. We then generate the quality-aware feature representations using t-SNE (Van der Maaten & Hinton, 2008), which are depicted in Fig. 5 (a) and (b). Fig. 5(b) presents the embedding space obtained when quality tokens are utilized and compared with the quality space without quality tokens (Fig. 5(a)). The overlapping part of the red area represents those features that are difficult for the model to distinguish. Our analysis reveals that our CSIQA is more discriminative for the quality features after introducing the quality token. To further demonstrate the unique features modeled by different tokens (for cls token and quality token), we compute the cosine similarity between the class token and quality token (averaged over the LIVEC and KonIQ datasets) in Fig. 5(c) and (d), resulting in low values of 0.29 and 0.069, which is significantly lower than the similarity between class and distillation labels in previous work (Touvron et al., 2021); 0.96 and 0.94 in DeiT-T and Deit-S, respectively (Naseer et al., 2021). This finding supports our hypothesis that the use of individual tokens for partial patch features in ViT can enable judges of the image quality from a different perspective (i.e., global and local).

**Ablation about logit distillation.** Fig. 6 presents the average SRCC with respect to the number of epochs on the LIVEC and KonIQ test sets. The results indicate that our CSIQA method exhibits remarkably faster convergence compared to other methods (Qin et al., 2023; Su et al., 2020; Zhang et al., 2018). Additionally, incorporating the logit distillation module in the training process results in an accelerated convergence speed and a further improvement in performance on both the LIVEC and KonIQ datasets (Compare CSIQA and w/o KD in Fig. 6). Notably, it achieves the fastest convergence after only a single epoch of training, outperforming the second-best NR-IQA method mentioned in the Tab. 1 of the main paper.

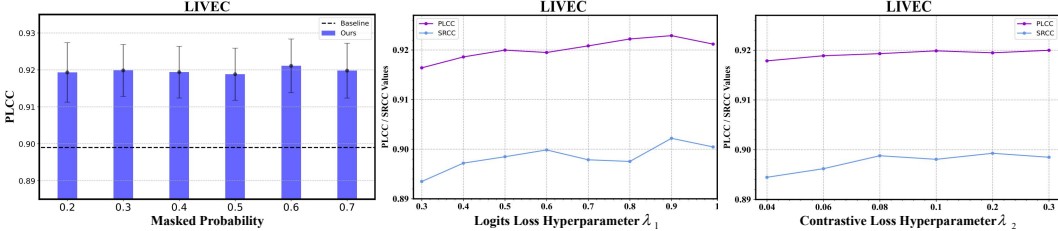

Figure 7: (a) Sensitivity analysis of patch mask probabilities (b) Hyperparameter $\lambda_1$ for balanced logit distillation loss (c) Hyperparameter $\lambda_2$ for balanced contrastive loss. All these results were obtained on the LIVEC dataset.

**Ablation about decoder.** We introduce a Transformer decoder for further improved feature refinement. It uses masked self-attention and cross-attention on "cls" and "qua" tokens. Masked self-attention minimizes irrelevant features, while cross-attention enhances quality-related aspects, adapting tokens for BIQA. The ablation experiments presented in Tab. 9 further validate the efficacy of the decoder in refining features.

Table 9: Ablation experiments concerning the decoder.

| Method | LIVE | | LIVEC | |
| --- | --- | --- | --- | --- |
| | PLCC | SRCC | PLCC | SRCC |
| VIT | 0.965 | 0.949 | 0.861 | 0.845 |
| CSIQA w/o decoder | 0.981 | 0.980 | 0.901 | 0.878 |
| CSIQA | 0.985(+0.4%) | 0.983(+0.3%) | 0.920(+1.9%) | 0.898(+2.0%) |

**Ablation about the hard-easy sampling strategy.** Tab. 10 outlines fine-tuning outcomes using both hard and hard-easy samples. When fine-tuning with simple examples, there's a slight performance drop, possibly due to model overfitting caused by an abundance of simple examples, and fine-tuning with hard samples yields 0.6% enhancement on the LIVEC dataset. we achieve a 1.3% increase in performance using easy-hard fine-tuning. Within our CSIQA framework (Tab. 11), using easy-hard sampling yields a noticeable enhancement (1.1%) compared to hard sampling. Easy-Hard Sampling helps us circumvent the issue of excessive hard samples leading to local minima, while also addressing overfitting stemming from straightforward examples.

Table 10: Ablation study about hard-easy sampling strategy for ResNet-50

| Sampling | Method | LIVE | | LIVEC | |
|---|---|---|---|---|---|
| | | PLCC | SRCC | PLCC | SRCC |
| | ResNet-50 | 0.942 | 0.938 | 0.858 | 0.851 |
| Easy | ResNet-50 | 0.949 | 0.945 | 0.857 | 0.848 |
| Hard | ResNet-50 | 0.952(+1.0%) | 0.947(+0.9%) | 0.864(+0.6%) | 0.858(+0.7%) |
| Easy-Hard | ResNet-50 | 0.956(+1.4%) | 0.950(+1.2%) | 0.871(+1.3%) | 0.858(+0.7%) |

Table 11: Ablation study about hard-easy sampling strategy for CSIQA

| Sampling | Method | (TID) PLCC | SRCC | (LIVEC) PLCC | SRCC |
|---|---|---|---|---|---|
| Hard | CSIQA | 0.929 | 0.914 | 0.900 | 0.876 |
| Easy-Hard | CSIQA | 0.939(+1.0%) | 0.925(+1.1%) | 0.910(+1.0%) | 0.889(+1.2%) |

**More comparisons with CONTRIQUE.** To make a fair comparison with CONTRIQUE (Madhusudana et al., 2022), we took the same ResNet-50 as our backbone. For further alignment, we tried two training methods. One is to fine-tune only the last encoder layer for contrastive learning and the fully connected layer (aligned with CONTRIQUE), and the other is to fine-tune it all. The experimental results are shown in Tab. 12. In the case of the frozen encoder, the method based on context contrastive learning has a 0.7% (1.1% fine-tuning) improvement over CONTRIQUE on the synthetic dataset LIVE. It is worth noting that on the real dataset LIVEC, That's a significant increase of 2.6% (2.5% fine-tuning). Because CONTRIQUE treats each distorted image as a unique class, it discusses the positive class contrast information that other real images of similar quality can provide. This indirectly reduces the difference perception of the model for real distorted images. However, our method can make up for this defect based on accurate and reliable labels, and can explicitly mine the subtle differences between images with different quality scores.

Table 12: Comparison of CSIQA with CONTRIQUE using ResNet-50 as the backbone, Co. denotes CONTRIQUE.

| Method | Backbone | LIVE | | LIVEC | |
|---|---|---|---|---|---|
| | | PLCC | SRCC | PLCC | SRCC |
| Baseline | ResNet-50 | 0.947 | 0.923 | 0.852 | 0.827 |
| Co.(Linear Regression) | ResNet-50 | 0.961 | 0.960 | 0.857 | 0.845 |
| CSIQA(Linear Regression) | ResNet-50 | 0.968(+0.7%) | 0.966(+0.6%) | 0.883(+2.6%) | 0.855(+1.0%) |
| CSIQA(fine-tuning) | ResNet-50 | 0.972(+1.1%) | 0.970(+1.0%) | 0.882(+2.5%) | 0.859(+1.4%) |

**More experimental results on the framework portability.** We assess existing IQA methods' generality, denoted by "+," signifying our model with our learning strategy. Pseudo-labels are from original pre-trained IQA models. Reported results show our method effectively enhances IQA networks. As evidenced by Tab. 13 and Tab. 14, our proposed model seamlessly integrates with existing IQA methodologies, resulting in notable performance enhancements.

**Experimental support for Proposition A.2.** As shown in Tab. 15 and Tab. 16, to illustrate contrastive learning's noise-resilience, we conducted experiments using ResNet-50 and VGG-16 backbones in

Table 13: We assess existing IQA methods' generality, denoted by "+" signifying our model with our learning strategy. Pseudo-labels are from original pre-trained IQA models. Reported results show our method effectively enhances IQA networks.

| Method | LIVE | | LIVEC | |
| --- | --- | --- | --- | --- |
| | PLCC | SRCC | PLCC | SRCC |
| DBCNN | 0.957 | 0.955 | 0.859 | 0.833 |
| DBCNN + | $0.970_{+1.3\%}$ | $0.968_{+1.3\%}$ | $0.878_{+1.9\%}$ | $0.859_{+2.6\%}$ |
| HyperNet | 0.966 | 0.962 | 0.880 | 0.867 |
| HyperNet + | $0.974_{+0.8\%}$ | $0.971_{+0.9\%}$ | $0.903_{+2.3\%}$ | $0.884_{+1.7\%}$ |
| TReS | 0.968 | 0.969 | 0.866 | 0.848 |
| TReS + | - | - | $0.887_{+2.2\%}$ | $0.865_{+1.7\%}$ |

Table 14: Context contrastive learning paradigm is applied to the IQA network ADANet (Pan et al., 2022) with the backbone of EfficientNet-b0 (Tan & Le, 2019) and RedNet101 (Li et al., 2021) (referred by "Effi-b0" and "Red101", respectively)."+" refers to the model combined with our learning strategy.

| Method | CSIQ | | LIVEC | | KonIQ | |
| --- | --- | --- | --- | --- | --- | --- |
| | PLCC | SRCC | PLCC | SRCC | PLCC | SRCC |
| Effi-b0 | 0.922 | 0.910 | 0.851 | 0.828 | 0.921 | 0.905 |
| Effi-b0 + | $0.932_{+1.0\%}$ | $0.920_{+1.0\%}$ | $0.865_{+1.4\%}$ | $0.849_{+2.1\%}$ | $0.928_{+0.7\%}$ | $0.913_{+0.8\%}$ |
| Red101 | 0.942 | 0.931 | 0.757 | 0.734 | 0.881 | 0.877 |
| Red101 + | $0.949_{+0.7\%}$ | $0.940_{+0.9\%}$ | $0.798_{+4.2\%}$ | $0.782_{+4.8\%}$ | $0.890_{+0.9\%}$ | $0.887_{+1.0\%}$ |

both contrastive and ranking settings across diverse datasets. The choice of comparison samples for both methods was influenced by noisy labels. Results summarized in the table highlight contrastive learning's competitive advantage. With the VGG-16 backbone, we improved by 8.1% on TID2013 and remained competitive on LIVE. Using RankNet-50, contrastive learning boosted performance by 5.3% on Kadid and 2.7% on KonIQ. This is attributed to contrastive learning's robustness in handling extreme sample noise. While ranking learning can be vulnerable to noise, especially when samples are closely ranked, contrastive learning, as demonstrated theoretically, sidesteps this issue. The ample size of 'n' in the theoretical proof allows the largest negative sample to converge toward its expectation and the smaller positive sample to approximate its positive counterpart. Consequently, contrastive learning exhibits heightened noise robustness.

Table 15: Comparison between contrastive learning and rank learning using the VGG16 backbone.

| Method | Backbone | LIVE | | TID2013 | |
| --- | --- | --- | --- | --- | --- |
| | | PLCC | SRCC | PLCC | SRCC |
| Learning to Rank | VGG16 | 0.976(+0.3%) | 0.975(+0.4%) | 0.803 | 0.791 |
| Contrast Learning | VGG16 | 0.973 | 0.971 | 0.884(+8.1%) | 0.863(+7.2%) |

# E  SENSITIVITY STUDY OF HYPERPARAMETERS

**Patch Masked Probability.** The probability of the random mask is a hyperparameter used to adjust the probability of the random quality-aware mask attention citation. We conduct ablation experiments with different random mask probabilities to examine their impact on CSIQA. Fig. 7(a) displays the variance bar chart, which reveals that our model is not particularly sensitive to changes in patch mask probability. However, appropriate occlusion can significantly enhance the model's stability (e.g., a smaller variance when the probability is 0.4 than when it is 0.2).

Table 16: Comparison between contrastive learning and rank learning using the ResNet backbone.

| Method | Backbone | KADID | | KonIQ | |
|---|---|---|---|---|---|
| | | PLCC | SRCC | PLCC | SRCC |
| Learning to Rank | ResNet-50 | 0.822 | 0.820 | 0.896 | 0.884 |
| Contrast Learning | ResNet-50 | 0.875(+5.3%) | 0.881(+6.1%) | 0.923(+2.7%) | 0.910(+2.6%) |

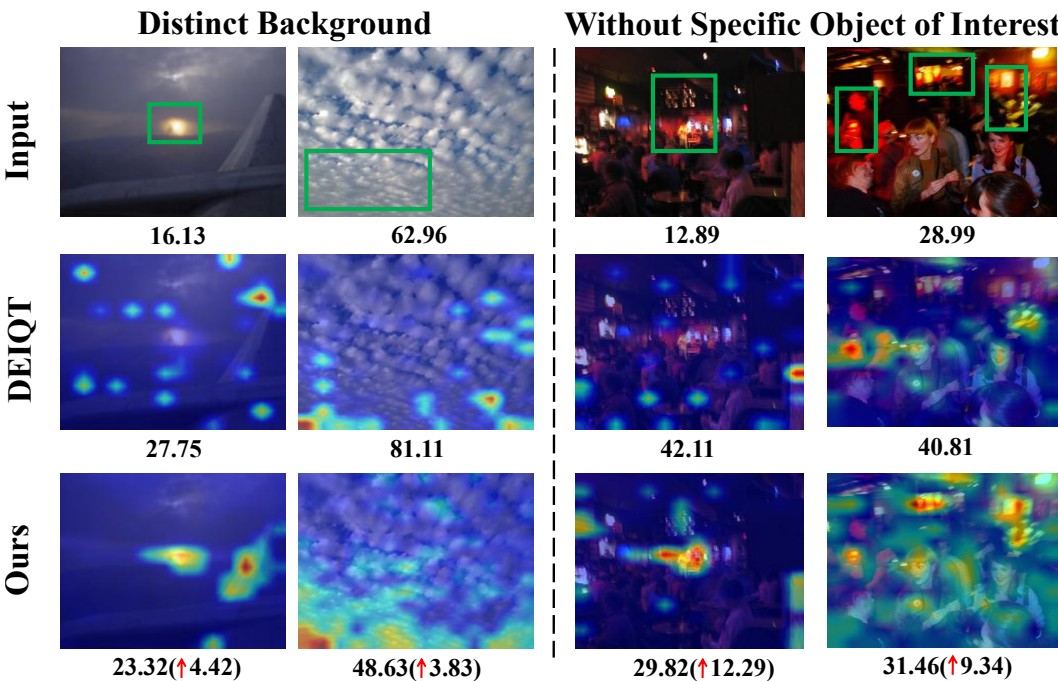

Figure 8: Activation maps of DEIQT (Qin et al., 2023) and CSIQA(ours) using Grad-CAM (Selvaraju et al., 2017). Scores in this figure represent *Mean Opinion Scores* (MOS). CSIQA still well-performs on background or without any specific object of interest, resulting in accurate prediction. Rows 1-3 represent input images, CAMs from DEIQT and CSIQA, respectively.

**Loss Weight Hyperparameter.** In this paper, we utilize $\lambda_1$ and $\lambda_2$ to balance logit supervision and contrastive learning. In this subsection, we conduct a sensitivity study of the hyperparameters to explore their effects. As shown in Fig. 7 (b) and (c), CSIQA is found to be sensitive to the loss weight $\lambda_1$, while less sensitive to the loss weight $\lambda_2$. Specifically, small values of $\lambda_1$ weaken the impact of our logit distillation, while large values of $\lambda_1$ may result in learning too many noisy pseudo-labels and lead to performance degradation. Therefore, choosing an appropriate value of $\lambda_1$(e.g., 0.9) is critical for achieving optimal performance. On the other hand, our experiments show that CSIQA is less sensitive to $\lambda_2$. However, it still plays an essential role in the overall performance of our model. We recommend using a moderate value of $\lambda_2$(e.g., 0.1) to achieve a balance between contrastive learning and logit supervision.

# F   QUALITATIVE ANALYSIS

**Visualization of activation map.** In IQA datasets, there tend to be a large number of images with distinct foreground/background or without any specific object of interest  (Sheikh et al., 2006; Ghadiyaram & Bovik, 2015). In the main paper, we show visualization results that show that our CSIQA can well focus on the distortion of the foreground part, as this is also more in line with the feature that the Human Visual System is significantly more concerned about (Ponomarenko et al., 2015; Hosu et al., 2020). In this section, we further visualize attention maps in images with distinct backgrounds or without any specific object of interest to embody the comprehensive ability of our

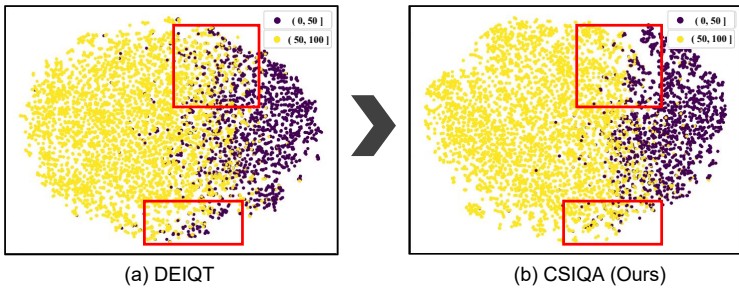

(a) DEIQT    (b) CSIQA (Ours)

Figure 9: The t-SNE of the current SOTA method DEIQT and our CSIQA visualizes 10073(all images) quality-aware representations learned from KonIQ.

algorithm to capture image distortion. As shown in Fig. 8, compared to DEIQT (the second best method in the main paper's Tab.1), our CSIQA shows the superior ability to accurately focus distinct backgrounds or lacking any specific object image distortion regions which are highlighted in the green box (i.e., blur sun, dim clouds, and overly bright red lights), resulting in improved prediction.

**Visualization of Feature Distributions across Different quality scores.** In Fig. 9, we further use t-SNE (Van der Maaten & Hinton, 2008)to visualize the feature distribution of CSIQA and DEIQT (Qin et al., 2023), across all images of KonIQ (a total of 10073 authentic images). The purple points represent lower-quality images (scoring less than 50 on a scale of 0-100), while the yellow points represent higher-quality images (scoring more than 50 on a scale of 0-100). As shown in Fig. 9(b), CSIQA demonstrates superior discriminative capability in distinguishing between high-quality and low-quality images. Specifically, the red box area highlights CSIQA's ability to accurately capture the distinguishing features of images with varying quality levels. Moreover, the intra-class distance between images of the same quality level is noticeably more compact with CSIQA, indicating its effectiveness in producing more distinct feature representations.

**Visualization of gMAD** The concept of gMAD involves a dual-model system comprising a defender and an attacker. The objective is to identify a pair of images for which the defender perceives similar quality levels, while the attacker discerns significant quality differences. Subsequent human evaluations are employed to adjudicate the correctness between the two models. This methodology underscores the proficiency of our model in both offensive and defensive capacities. As depicted in Fig. 10, our model excels in both offensive and defensive capacities, with its predicted image quality aligning closely with human evaluations.

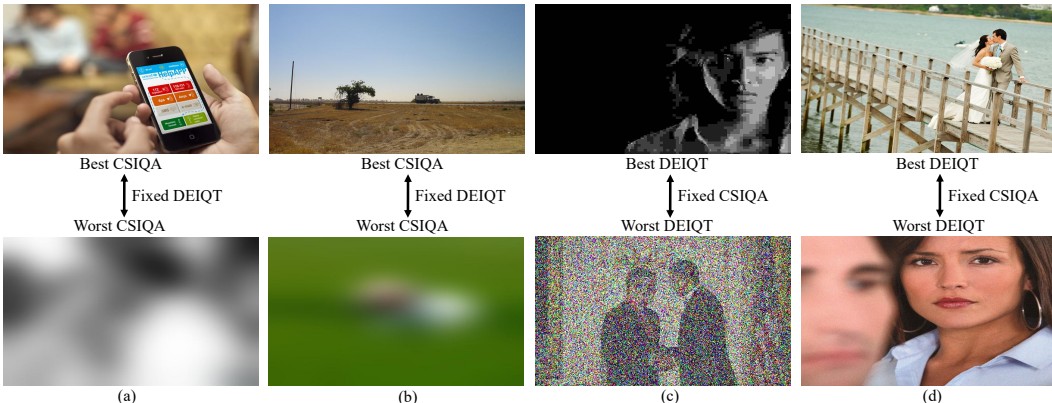

Figure 10: gMAD competition results. (a) Fixed DEIQT at low quality. (b) Fixed DEIQT at high quality. (c) Fixed CSIQA at low quality. (d) Fixed CSIQA at high quality.

