# OpenReview forum: "Perceptual Context and Sensitivity in Image Quality Assessment: A Human-Centric Approach"
_ICLR.cc/2024/Conference — ICLR 2024 Conference Withdrawn Submission_

### Official Review · Reviewer_GoQM · 2023-10-24

**Soundness:** 1 poor
**Presentation:** 1 poor
**Contribution:** 1 poor
**Rating:** 1
**Confidence:** 5

**Summary:**

This work proposes a BIQA model that is designed to emulate the perceptual mechanisms inherent to the Human Visual System (HVS) for the purpose of evaluating image quality. Specifically, the proposed method is composed of two components: 1) A quality-aware contrastive learning module; 2) A random mask mechanism. Experiemnts are conducted on 8 popular IQA databases.

**Strengths:**

+ Rich experimental results are reported.

**Weaknesses:**

- The novelty of this work is limited for the following two reasons:

First, contrastive learning has been extensively studied in the field of BIQA, except for the mentioned CONTRIQUE, at least two existing works are proposed in early 2023 [1,2].

[1] Saha, A., Mishra, S., & Bovik, A. C. (2023). Re-IQA: Unsupervised Learning for Image Quality Assessment in the Wild. In Proceedings of the IEEE/CVF Conference on Computer Vision and Pattern Recognition (pp. 5846-5855).

[2] Zhao, K., Yuan, K., Sun, M., Li, M., & Wen, X. (2023). Quality-aware pre-trained models for blind image quality assessment. In Proceedings of the IEEE/CVF Conference on Computer Vision and Pattern Recognition (pp. 22302-22313).

In these previous works, the underlying motivation is to leverage unsupervised contrastive learning to enable BIQA model training on much larger data beyond the available IQA datasets with human quality annotations. This paper arrives at a similar approach from a seemingly different perspective. As stated in Sec. 3.3, this paper tries to use contrastive learning to better utilize the label information, resulting in a supervised solution. The explanation of this practice is very far-fetched, and the theoretical analysis is not rigorous either. In Theorem 2, the formulation of rank learning is not correct. Consequently, I think the use of supervised contrastive learning here is just doing something for the sake of doing it.

Second, the mask mechanism is a trivial application of MAE [3], but it is not even referenced in the manuscript. What makes things even worse is that the authors attempt to forcefully interpret this as a mechanism inspired by the HVS.

[3] He, K., Chen, X., Xie, S., Li, Y., Dollár, P., & Girshick, R. (2022). Masked autoencoders are scalable vision learners. In Proceedings of the IEEE/CVF conference on computer vision and pattern recognition (pp. 16000-16009).

- The combination of knowledge distillation and ground-truth labels is fundamentally flawed. If the pseudo labels generated by the teacher model are not accurate enough, this will definitely pollute the training process. However, if we already have a very strong teacher model, why should we additionally train a student model using such a complex scheme? For model compression? Apparently not in this work.

- SRCC and PLCC range from -1 to 1, instead of 0 to 1.

- It is very difficult to follow the definition of the hard-easy samples part.

- The overall framework is an extremely entangled thing, from which I do not think the community can learn anything.

**Questions:**

It is unclear how the authors report the results of other methods in Table 1. Do all of them have the same training/val/test splits?

---

### Official Review · Reviewer_8aSJ · 2023-10-30

**Soundness:** 2 fair
**Presentation:** 2 fair
**Contribution:** 2 fair
**Rating:** 3
**Confidence:** 5

**Summary:**

A contrastive learning approach is used to develop BIQA models.

**Strengths:**

1. The performance seems to look great.
2. The cross-dataset evaluation and the adoption of gMAD competition to verify the generalization is a plus.

**Weaknesses:**

1. The proposed method may be less motivated. For "context," more fine-grained contrastive relationships between different quality samples have been extensively investigated in the field of IQA, under the learning to ranking formulation and starting from Gao et al.'s paper back to 2015. For  "Sensitivity," the authors do not provide empirical evidence that ImageNet initialization + finetuning is weaker than their contrastive learning setting. The better performance of the proposed method may simply stem from the fact that a more powerful network architecture with much larger model parameters is adopted.

2. The authors argue that ImageNet-like initialization is not good but start a pre-trained encoder based on ViT-S.

3. Both the teacher and the student networks are trained. Which one is adopted for testing?

4. The details of gMAD competition seem to be vague. Fig. 3 and Fig. 10 compare two models with the same setting, but some of the gMAD pairs are different.

5. For each computational module, there are several hyperparameters, e.g., $\tau$ in Eq. (4), $\gamma_1$, and $\gamma_2$ in Eq. (6). How do these affect the final performance?

**Questions:**

1. Section 3.5 seems less relevant.

2. 8:2 training: test set splitting without validation set may be an issue.

**Details Of Ethics Concerns:**

N.A.

---

### Official Review · Reviewer_xjPV · 2023-10-31

**Soundness:** 3 good
**Presentation:** 2 fair
**Contribution:** 3 good
**Rating:** 6
**Confidence:** 4

**Summary:**

This paper proposed a Quality Context Contrastive Learning module to capture potential quality correlations in the global context of the dataset and a Quality-aware mask attention module to improve the model’s perception of local distortions. Some experiments are conducted to demonstrate the effectiveness of the proposed method.

**Strengths:**

1.	This paper proposes a novel BIQA framework and achieves state-of-the-art performance.
2.	Extensive and sufficient experimental analysis demonstrates the effectiveness of the proposed methods

**Weaknesses:**

1.	Lack of sufficient review of recent related methods.
2.	The writing of the paper is not clear enough and not easy to understand.

**Questions:**

1.	CVRKD-IQA [1] and UNIQUE [2] also learn image quality representations through the process of comparing the quality of different images. It is better to review them.
[1] Yin G, Wang W, Yuan Z, et al. Content-variant reference image quality assessment via knowledge distillation[C]//Proceedings of the AAAI Conference on Artificial Intelligence. 2022, 36(3): 3134-3142.
[2] Zhang W, Ma K, Zhai G, et al. Uncertainty-aware blind image quality assessment in the laboratory and wild[J]. IEEE Transactions on Image Processing, 2021, 30: 3474-3486.
2.	The relationship between the part of knowledge distillation and the other lacks clarity. The authors may need to be added to some diagrams or descriptions.
3.	How does the easy-hard sample mitigate overfitting? Can further explanation be given? Is there any relevant basis?
4.	What dataset was the teacher network pre-trained on? Is it the same data used for training the student network? As well as the fact that normally the teacher network is supposed to have better performance than the student network, it is clear in this paper that the student network is stronger than the teacher network, so the addition of the logit loss is supposed to reduce the performance. The authors may need further clarification on this point.
5.	In the Ablation study, comparing index c) and d), index f) and g), and index i) and k), the performance gain from MA is almost negligible. Besides, there is a writing error: “we compare index b) and c)” -> “we compare index a) and b)”
6.	It can be seen in Fig. 4, that although BEIQT is weaker than the proposed algorithm in MSE, the ranking is more accurate compared to the proposed method (the proposed method is incorrect in ranking the quality of image 2 and image 3).

---

### Official Review · Reviewer_GAJw · 2023-11-06

**Soundness:** 2 fair
**Presentation:** 3 good
**Contribution:** 2 fair
**Rating:** 5
**Confidence:** 5

**Summary:**

The paper introduces CSIQA ( Perceptual Context and Sensitivity in Blind Image Quality Assessment Algorithm).  The work aims to look into image quality from both a local and global context. They introduce a novel quality-aware mask attention and use a quality-aware contrastive learning module in this paper. The paper is well organized & written and complete with a substantial number of experiments/ ablation studies.

**Strengths:**

1. The paper introduces the novel quality-aware mask attention.
2. Usage of distillation strategy to improve training
3. First paper in IQA leveraging contrastive learning that tries to define hard vs easy positive and negative samples and study its effect on the training process.
4. Decent attempt at providing mathematical proofs/intuitions behind the CSIQA design

**Weaknesses:**

Comment on Paper Introduction :
Figure 1 of the paper is misleading with respect to the state-of-the-art IQA methods in 2023/2024.  It is borderline incorrect to argue that current state-of-the-art methods in 2023/2024 use transfer learning using pre-trained ImageNet models. In 2021, we could agree, but not in 2023. I think the authors should cite the current state of the methods that employ contrastive/self-supervised learning (exemplar methods [1],[2],[3]) and discuss how each of the existing techniques differs from the proposed method clearly describing the shortcomings and issues addressed in this paper.  This will help readers better understand how this work is novel/different with respect to the existing results employing self-supervised learning. The authors should change Fig 1 as it is misleading with respect to the state-of-the-art in 2023/2024.

Comment on Contribution :
#1 [1],[2]. [3] methods are already comparing across images, so broadly saying that existing methods have this as a limitation is untrue.
Instead, the authors should argue that their networks were pretrained only on the ImageNet and not any other extra specific datasets used in [2],[3].

Comment on Related Work
[3] attempts to contrast based on local patch quality and is very similar to the proposed method; hence, it should be discussed in a related section on IQA with a local perspective. [4] another popular IQA method that studies local to global image quality needs to be discussed.

Comment on Results :
The results show superior performance of the CSIQA framework. However, the results shown are very similar to some existing methods, especially DEIQT. I think authors should conduct t-tests to verify if the differences in the results are statistically significant. Otherwise, it is difficult to conclude if this method performs better than existing methods. Authors are requested to conduct t-tests to verify the results are statistically superior as compared to DEIQT, [1], [2] and [3] and report them in the paper. Also, a performance vs complexity trade-off graph should be shown along with the results. Methods like [2], and [3] are ResNet-50 based, so the authors should compare objective results with the complexity of the proposed methods with other compared methods. Results in Table 12 attempt to compare the results of [2] with CSIQA. However, the results are pretty close, and again, conducting t-tests is needed to verify the statistical superiority of CSIQA.


[1] Quality-Aware Pre-Trained Models for Blind Image Quality Assessment, Zhao et al. CVPR 2023

[2] Image Quality Assessment  Using Contrastive Learning, Madhusudana et al., 2022 IEEE TIP 2022

[3] Re-IQA: Unsupervised Learning for Image Quality Assessment in the Wild, Saha et al. CVPR 2023

[4] From Patches to Pictures (PaQ-2-PiQ): Mapping the Perceptual Space of Picture Quality, Ying et al. CVPR 2020

**Questions:**

Question on Positive/Negative Sampling Strategy :

The positive and negative examples obtained are based on some fractions (<20% vs >60%). Consider an anchor image I and another image J; the images I & J can be positive and negative samples based on the other images in the batch. Please explain how this inconsistency should be regarded as a valid sampling strategy.

The same holds for easy vs complex example sampling.

---

### Official Review · Reviewer_NFhk · 2023-11-10

**Soundness:** 2 fair
**Presentation:** 3 good
**Contribution:** 2 fair
**Rating:** 5
**Confidence:** 5

**Summary:**

The paper proposed a novel image quality assessment method. It relies on the use of contrastive learning for better emulating HVS proficiency in comparing image quality. To mitigate overfitting, it exploits an hard negative mining strategy within a knowledge distillation framework. Furthermore, a patch-level masking strategy is applied to enforce the model focusing on local distortions rather than semantic information. Experiments on eight benchmark datasets demonstrate the effectiveness of the proposed method.

**Strengths:**

+ The paper is well written and organized.
+ The achieved performance highlight the effectiveness of the proposed method.

**Weaknesses:**

- The proposed method lacks innovation, and it is simply the combination of existing approaches (token masking, model distilling, hard negative mining, and contrastive learning).
- The manuscript lacks an explanation of how various design choices improve the representation obtained peculiarly for IQA.
- The improvement in correlation is marginal compared with the first SOTA method (i.e. DEIQT).

**Questions:**

- How does the easy-hard sample mitigate overfitting? Can the authors provide further details and explanations?
- Why do authors change the batch size according to the cardinality of the dataset? Above all, what sense does it make to decrease the batch size without complementarily decreasing the learning rate?
- Are the authors sure about the correlation range [0,1]? Is not possible to obtain negative correlation for Pearson and Spearman?

**Details Of Ethics Concerns:**

I believe this submission does not require any further specialized ethics review.